# Regional Pollination Activity by Moth Migration in *Athetis lepigone*

**DOI:** 10.3390/plants12193406

**Published:** 2023-09-27

**Authors:** Huiru Jia, Yuchao Chen, Xiaokang Li, Yunfei Pan, Dazhong Liu, Yongqiang Liu, Kongming Wu

**Affiliations:** 1State Key Laboratory for Biology of Plant Diseases and Insect Pests, Institute of Plant Protection, Chinese Academy of Agricultural Sciences, Beijing 100193, China; jhuiru@163.com (H.J.); 15978356769@163.com (Y.C.); lixiaokang2016@163.com (X.L.); panyunfei@hotmail.com (Y.P.); liudazhong94@163.com (D.L.); lyq364467268@163.com (Y.L.); 2Guangdong Laboratory for Lingnan Modern Agriculture, Guangzhou 510640, China; 3College of Plant Protection, Shenyang Agricultural University, Shenyang 110866, China

**Keywords:** noctuid moth, pollen study, nocturnal pollinators, migration routes

## Abstract

Nocturnal moths (Lepidoptera) are important pollinators of a wide range of plant species. Understanding the foraging preferences of these insects is essential for their scientific management. However, this information is lacking for most moth species. The present study was therefore conducted to delineate the host plant feeding and pollination ranges of an agriculturally important nocturnal moth species *Athetis lepigone* by identifying the pollen species adhering to their bodies during long-distance migration. Pollen grains were dislodged from 1871 *A. lepigone* migrants captured on Beihuang Island in the Bohai Strait between 2020 and 2021. This region is a key seasonal migration pathway for *A. lepigone* in northern China. Almost 20% of all moths sampled harbored pollens, providing direct evidences that this moth species may serve as pollinators. Moreover, at least 39 pollen taxa spanning 21 plant families and 31 genera were identified, with a preference for Asteraceae, Amaranthaceae, and Pinaceae. Additionally, the pollen adherence ratios and taxa varied with moth sex, inter-annual changes, and seasonal fluctuations. Most importantly, the pollen taxa were correlated with insect migration stages and indicated that *A. lepigone* bidirectionally migrates between central China (Shandong, Hebei, and Henan Provinces) and northeastern China (Liaoning Province). Overall, the findings of the present work provide valuable information on the pollination behavior, geographical origins, and pollination regions of *A. lepigone* moths and could facilitate the design and optimization of efficacious local and regional management strategies for this important insect.

## 1. Introduction

Plants and insects have evolved symbiotic relationships that play pivotal roles in ecosystem functionality and the sustenance of global agriculture [1,2,3]. One such relationship is pollination, a process wherein insects, particularly flower-visiting ones, assist in the reproduction of plants [4,5]. While the importance of bees in this process is well-documented, nocturnal moths (Lepidoptera), comprising over 40,000 identified species globally [6,7], also play a significant role in pollinating various plant species across different ecosystems [8,9,10,11,12].

Understanding the behavior and dietary preferences of these moths is essential, not just because of their pollination role, but also due to their potential as agricultural pests. For instance, while adult nocturnal moths feed primarily on nectar and pollen, their larvae exhibit herbivorous tendencies, causing damage to host plants [13]. Gaining knowledge about these plants is pivotal for effective pest management.

To delve deeper into the dietary preferences of nocturnal moths, researchers have turned to novel techniques like examining pollen grains on their bodies. This is based on the understanding that flower-visiting insects inadvertently gather and transport pollen from one flower to another [14,15,16,17]. Over the past few decades, this method has successfully disclosed the plants visited and consumed by various insects in general and by nocturnal moths in particular [18,19,20,21]. Advances in this field, including high-resolution scanning electron microscopy (SEM) [22], plant DNA barcoding [23], and high-throughput sequencing [24,25], have made pollen identification more precise.

*Athetis lepigone* Möschler (Lepidoptera: Noctuidae) serves as a pertinent example. This moth is an agriculturally important moth species with high migration capacity and fecundity [26,27]. Although native to many European and Asian countries, its recognition as an agricultural pest was first reported in 2005, following considerable damage to maize crops in Hebei Province, China [28,29]. Its detrimental impact has only increased over the years, spreading to multiple provinces and severely affecting maize yields [30,31]. Previous studies have shown that while *A. lepigone* larvae can feed on a diverse range of host plants, they exhibit a preference for maize [32,33]. However, the extent and significance of the pollination services offered by adult *A. lepigone* moths remain largely unexplored.

Our study aims to bridge this knowledge gap. By analyzing pollen grains found on migrating adult *A. lepigone* moths, we intend to deduce their adult host plant range. Our research focuses on specimens from Beihuang Island, situated in the Bohai Strait—a recognized migration pathway for this species (Figure 1) [27]. Leveraging techniques, such as SEM and ITS2-based plant barcoding, we identify the pollen species on the sampled moths, subsequently defining the potential pollination areas of *A. lepigone* based on the geographical distribution of these plants.

## 2. Results

### 2.1. Plant Hosts Inferred from Pollen Grains

We collected 1871 samples of migratory *A. lepigone* moths (771 females and 1100 males) at Beihuang Island during the experimental period of 2020–2021. Of these, 373 (19.94%) had pollen adhering mainly to their proboscises. The pollen was almost always detected in great abundance. Hence, the moths acquired the pollen through active contact during feeding rather than incidental contact with wind-borne pollen grains.

Analyses of the external morphology, size, and shape of the pollen grains on the 373 *A. lepigone* individuals revealed 39 distinct taxa (Figure 2). The pollen grains ranged from spherical to elliptical to triangular. All except two species of pollen grains were monads, and they ranged from very large [(*Cynanchum acutum* (>500 µm) and *Vincetoxicum* L. (>300 µm)] to very small (Asteraceae; 10–15 µm) in diameter. About 90% (35/39) of the pollen grains were medium or small in size (26–50 µm or 10–25 µm, respectively). The surface features of pollen grains are the most crucial morphological traits for species identification purposes. The pollen surface sculpturing patterns observed here included smooth (psilate), netlike (reticulate), ball of string-like (striate), or spinelike (echinate).

The various pollen types were also subjected to molecular confirmation via the *ITS2* locus. This procedure differentiated 34/39 pollen types beyond the genus level, and there was 99–100% similarity in sequence data and coverage. Figure 2 shows that 23/39 (58.9%) of the pollen grains were identified at the species level and included *Morus alba*, *Prunus avium*, *Amorpha fruticosa*, *Tamarix chinensis*, *Robinia pseudacacia*, *Ailanthus altissima*, *Torilis arvensis*, *Sorghum bicolor*, *Nicotiana tabacum*, *Cynanchum acutum*, *Amaranthus blitum*, *Cuscuta japonica*, *Humulus lupulus*, *Chenopodium album*, *Rubia cordifolia*, *Adenophora trachelioides*, *Suaeda glauca*, *Helianthus annuus*, *Tripolium vulgare*, *Ambrosia trifida*, *Aster tataricus, Eclipta prostrata*, and *Chrysanthemum indicum*. Fifteen pollen taxa were identified at the genus level and included *Vincetoxicum* L., *Salix* L., *Brassica* L., *Pinus* L., *Tilia* L., *Oenothera* L., and *Artemisia* spp. L(5). Overall, 39 pollen taxa from at least 22 and 31 plant families and genera, respectively, were identified. In most cases, each plant host was borne on a single moth. However, a few adult moths bore pollen originating from several different plant species.

### 2.2. Sex Differences in Pollen-Bearing Frequency

When the trial years were combined, there were significant differences between moth sexes in terms of the mean annual frequency of pollen occurrence on them. Pollen adhered to 26.33% (203/771) of all females and 15.45% (170/1100) of all males (χ2 = 33.5854; *p* = 6.8203 × 10^−9^). When the trial years were analyzed separately, however, the aforementioned difference was significant only in 2021 (χ2 = 64.4854; *p* = 9.7249 × 10^−16^) (Table 2).

### 2.3. Intra-Annual Shifts in Pollen Taxa

There were significant annual variations in the pollen species adhering to *A. lepigone*. Pollen species diversity widely varied during the research period. While 30 different pollen species were detected in 2020, only 24 were observed in 2021. The number of insects examined directly influenced both the number of pollen grains recovered and pollen taxa diversity. Therefore, pollen grain recovery and, by extension, pollen species diversity would both increase with sample size.

We also noted seasonal variations in the numbers and types of pollen species detected. We identified 19 species on the spring–summer (April–August) *A. lepigone* migrants. *Cynanchum acutum*, *Amorpha fruticose*, and *Chenopodium giganteum* were the most common and accounted for 52% of the total. We found 20 pollen species on the autumn *A. lepigone* migrants. The pollen from Asteraceae, such as *Chrysanthemum zawadskii*, *Artemisia* L., and *Ambrosia trifida*, were the most common and accounted for >62.2% of the total.

### 2.4. Characteristics of the Plants Producing the Pollen Collected by A. lepigone

The plants from which the adherent pollen grains originated included trees, shrubs, vines, and herbs. Only one type of pollen was derived from gymnosperms (*Pinus* L.). All other pollen types were produced by angiosperms (Figure 3). Of these, herbaceous plants outnumbered woody plants (χ2 = 112.26; df = 1; *p* < 0.001). Hence, *A. lepigone* preferentially visits herbaceous angiosperm flowers.

### 2.5. Possible Migration Trajectories of A. lepigone

Certain plants endemic to specific ecological zones and locations may serve to track long-distance insect migration (Jones and Jones, 2001). We correlated the foregoing findings with the migratory behavior of *A. lepigone* moths and their potential pollination areas inferred from the geographical distributions and flowering periods of their host plants.

No pollen from plants exclusive to the southern regions was detected on any moth during the spring–summer migration period. Pollen from *Tilia* L. was identified on the *A. lepigone* migrants sampled in June. Though *Tilia* L. is distributed in the northeast, it had not yet blossomed there as of June 10. In contrast, *Tilia* L. flowering reaches its peak in North China at that time. By that time, the *Tilia* L. flowering season had already ended in the south. Hence, the migrating moths originated mainly in North China and possibly in Hebei, Henan, and Shandong Provinces. Similar conclusions may be drawn for *Tamarix chinensis* and *Prunus avium* when their distributions and flowering periods are considered. The preceding results indicate that most of the summertime *A. lepigone* migrants originate in the North China region. During the autumn migration period, however, *Chrysanthemum zawadskii* and *Adenophora trachelioides* pollen grains were detected on *A. lepigone*. For this reason, these migrants were distributed mainly in Northeast China (Figure 4).

## 3. Discussion

Beihuang (BH) Island’s unique geographic position in the Bohai Gulf’s center—a key insect migration route—and specific ecological features, such as the absence of fresh water and arable land, making it an excellent location for studying insect migration in Northern China. Over the past two decades, this site served as a pivotal platform for deciphering the migratory behaviors of various insect species including *A. lepigone* [34]. Specifically, our previous studies conducted on this site, provided-to our best knowledge-the first direct evidence for the long-distance migration ability of *A. lepigone* by means of searchlight trapping, ovarian dissection and carbon isotope analysis [27]. It was established that this species seasonally migrates across the Bohai Sea. Here, we selected BH as our sampling site to collect migratory *A. lepigone* populations. We trapped potential migrants from April to October during the 2020–2021 research periods and found that their population dynamics were consistent with those of prior investigations. These results indicated that overseas *A. lepigone* migration has become a “regular ecological event”.

As 373/1871 (19.94%) of the migratory *A. lepigone* individuals, we inspected bore pollen grains on their bodies, this species is a potential pollinator. Moreover, this rate of pollen detection was comparable to those observed for other noctuid moths at the same site including *Spodoptera exigua* (16.1%) [35], *Agrotis segetum* (17.03%) [36]. Furthermore, the pollen loads for *A. lepigone* and other migratory insects at BH significantly varied on a diurnal, monthly, and annual basis. Prior research on *S. exigua* revealed that plant phenology, nectar viscosity, pollen grain characteristics, and migratory routes explain the observed variability in pollen loading [35]. Interestingly, we also found that the female moths had consistently higher overall, monthly, and annual pollen burdens than the males. The difference in body size between sexes may account for this discrepancy. In general, female moths have a bigger body size than male ones, leading to carry relatively more pollen grains and travel longer distances. A recent report by Földesi et al. (2020) shows that larger pollinators indeed deposit comparatively more pollen than smaller ones [37].

We analyzed the pollen grains on *A. lepigone* using an integrative approach combining traditional morphological methods (such as SEM) with ITS2 DNA barcoding [21,35]. This innovative methodology was developed in an attempt to overcome the limitations of each technique, morphological examination via scanning electron microscopy (SEM) and/or DNA barcoding. More specifically, SEM distinguishes pollen types according to their unique microscopic morphology. Nevertheless, this approach is labor-intensive, requires technical expertise, and may not effectively or reliably resolve the morphological similar pollen grains produced by closely related plant species [38]. In DNA barcoding, however, pollen DNA is extracted and analyzed. This method is relatively accurate as each type of pollen has its unique DNA profile. Nevertheless, DNA barcoding relies heavily upon reference databases such as GenBank that complicate identification when particular species reference sequences are missing or unavailable [39]. The integration of both methods mutually offsets their limitations. This novel approach has been successfully applied and reported in several recently published studies [20,21].

Our research team has prioritized the study of the pollen grains adhering to insect bodies. To date, we have analyzed the pollen gathered by various noctuids (see in pollen studies for *Spodoptera exigua* [35]). The present work used the aforementioned combined approach to analyze the pollen borne by *A. lepigone*. We discovered that adult *A. lepigone* moths gathered pollen grains from >39 distinct host plant species. It is important to note that our samples were exclusively sourced from a single location. Considering the inherent geographical specificity in plant distribution [21], we are confident that the true count of plant species frequented by *A. lepigone* surpasses what we have documented in this study. Furthermore, we found that nearly all pollen taxa detected were borne by one or more moth species regardless of the number of moths examined (see in pollen studies for *Spodoptera exigua* [35]),. Thus, there is a certain degree of homogeneity in terms of the types of flowers pollinated by nocturnal moths. Pollen grains from *Oenothera* L. and *Vincetoxicum* L. were found on the proboscis of almost all moth species explored here (see in pollen studies for *Spodoptera exigua* [35]). We recently discussed the possibility that a moth species exhibits a conspecific attraction for habitat selection [35]. This phenomenon has also been observed in fruit flies and butterflies. In these cases, flower-visiting insects from the same taxon tend to pollinate the same species or groups of plants. The foregoing findings underscore the necessity to conduct further investigations into this behavior.

Pollen grain analyses also help establish the geographical origins and migration routes of insects [14,17]. In recent decades, pollen has been used to analyze the migration patterns of moths [18,19,20,21,22,23,24,25,26,27,28,29]. Adult *Helicoverpa zea*, *Agrotis ipsilon*, and *Pseudaletia unipuncta* detected in the northern United States bore exotic pollen grains originating from Texas. Thus, the migration routes of the preceding moth species extended for several hundred kilometers [18,19]. By identifying the pollen of certain plant species (e.g., *Citrus sinensis*, *Flueggea virosa*, and *Melia azedarach*) adhering to several different insects, we demonstrated that the latter migrate from southern to northern China across the Bohai Sea in late spring and summer [35]. Regarding *A. lepigone*, direct empirical evidence for its migratory behavior have been obtained for many years [27], but the migration patterns for this moth species remained to be elucidated. To the best of our knowledge, the present study was one of the first to indicate that *A. lepigone* migrates northward in springtime and returns southward in autumn. This migratory pattern is consistent with those of other nocturnal moths previously studied at the same location and East Asian monsoon airflows. It is worth noting that, during the spring-summer migration periods, *A. lepigone* did not bear the same southern-specific pollen sources as those of other migratory nocturnal moths captured at the same location. Rather, *A. lepigone* carried pollen from plant species distributed mainly in North China including Shanxi, Henan, Shandong, Anhui, and Hebei Provinces. This discovery aligned with previous reports on the occurrence range of this moth in China [28,29]. From these data, we concluded that *A. lepigone* has a shorter migratory distance than certain other nocturnal moths. In order to obtain more precise migration path of *A. lepigone*, other tracking methods, such as internal tracing elements [40], trajectory analysis [41], and genetic diversity studies [42], could be used in the future.

## 4. Materials and Methods

### 4.1. Study Site and Sample Collection

Beihuang Island (BH; Shandong Province, China; 38°24′ N, 120°55′ E; ~2.5 km^2^) is located in the center of the Bohai Gulf, where is one prime (insect) migration corridor (Figure 1). As BH is remote from the mainland and lacks fresh water and arable land, it is suitable for the study of insect migration in northern China. Based on the geography of BH, it was assumed that all insects caught in light traps there were migratory and had to make a journey at high altitude to a distance of ≥40–60 km from the mainland [36,37].

A vertically oriented searchlight trap (No. DK.Z.J1000B/t; Shanghai Yaming Lighting Co., Ltd., Shanghai, China) fitted with a 1000-Wmetal halide-lamp (No. JLZ1000BT; Shanghai Yaming Lighting Co., Ltd.) [43] was used to sample insects on BH every night between April and October of 2020 and 2021. Either nightly subsamples of 20 *A. lepigone* moths or all available individuals were randomly selected, transferred to 2-mL tubes, and stored at −20 °C until further processing. In total, we sampled 1871 individuals comprising 771 females and 1100 males (with 1207 individuals sampled in 2020 and 664 in 2021).

### 4.2. Microscopic Pollen Examination

Earlier studies demonstrated that pollen may adhere to the antennae, eyes, or legs of moths. However, most of the pollen was localized to the proboscises [18,19]. Hence, the present study mainly focused on the proboscises of the *A. lepigone* moths sampled [35,36].

Light microscopy and scanning electron microscopy (SEM) were implemented in the present work according to the methods of Liu et al. (2016) [20]. Briefly, individual moth proboscises were carefully excised with non-magnetic tweezers, mounted on glass slides, and examined under a stereomicroscope (No. SZX16; Olympus Corp., Pittsburgh, PA, USA) at 200× magnification. Following this, the presumed pollen grains were gently detached from the proboscises, affixed to aluminum stubs with double-sided adhesive tape, coated with a thin layer of gold, and visualized either under a Hitachi Regulus 8100 ultra-high-resolution field-emission scanning electron microscope (Hitachi, Tokyo, Japan) in our laboratory or under a Hitachi S570 scanning electron microscope (Hitachi) at the Electronic Microscopy Centre of the Institute of Food Science and Technology (CAAS; Beijing, China).

### 4.3. Molecular Analyses

The pollen grains detected on the *A. lepigone* proboscises were then subjected to DNA barcoding. This methodology comprises DNA extraction from a single pollen grain, PCR amplification, cloning, and DNA sequencing.

DNA Extraction. Genomic DNA was extracted from individual pollen grains following the protocol of Chen et al. (2008) [44]. Each pollen grain was placed in a PCR tube containing 5 µL of lysis solution (0.1 M NaOH plus 2% (*v*/*v*) Tween^®^ 20) and heated in a GeneAmp PCR System 9700 (Applied Biosystems, Foster City, CA, USA) at 95 °C for 17.5 min. Then, 5 µL of Tris-EDTA (TE) buffer was added to the tube and the contents served as a template for the subsequent PCR amplifications.

PCR Amplification. The partial nuclear ribosomal internal transcribed spacer (nrITS2) region is a universal barcode marker for plant identification. Here, it was amplified using the forward and reverse plant-specific primers ITS-p3 and ITS4, respectively [45,46]. All PCRs were conducted in a 25-µL reaction volume consisting of 2 µL of DNA template, 0.5 μM each of the forward and reverse primers, and 12.5 µL of 2×high-fidelity *Platinum* SuperFi II Master Mix (Thermo Fisher Scientific, Waltham, MA, USA). The PCR cycle parameters included an initial denaturation step at 98 °C for 3 min followed by 35 cycles of 98 °C for 1 min, 60 °C for 30 s, and 72 °C for 1 min and a final elongation step at 72 °C for 1 min.

Cloning and DNA Sequencing. The PCR products were purified with an E.Z.N.A. Gel Extraction Kit (Omega Bio-tek, Norcross, GA, USA), and ligated into a pClone007 vector (Tsingke BioTech, Beijing, China). Five positive clones were then randomly selected and subjected to Sanger sequencing with M13 primers either at Sangon Biotech Co., Ltd. (Shanghai, China) or at Beijing Genomics Institute (BGI; Shenzhen, China).

### 4.4. Pollen and Host Plant Identification

Each pollen grain was classified according to its molecular characteristics, morphological traits, and geographic distribution. Firstly, a BLASTn online search was conducted to compare the genetic sequences against those stored in the National Center for Biotechnology Information (NCBI; https://www.ncbi.nlm.nih.gov/search/ accessed on 23 February 2023) database. If the top 5 bit scores for a sequence corresponded to a single species, multiple species within a genus, or multiple genera within a family, then the sequence was attributed to that species, genus, or family, respectively. Sequences aligning with multiple families were deemed unidentifiable [47] and several sequence taxa were classified at either the genus or the family level. Identifications based on molecular data were supplemented with morphological characterizations based on published SEM images of pollen grains from Chinese flora [48,49], online search engines, or palynological databases (https://www.paldat.org/). Species-level identifications were cross-referenced against the Flora of China Species Library (https://species.sciencereading.cn) and the Plant Science Data Center (https://www.plantplus.cn/cn) to confirm the presence of the host plant within the broader study area.

### 4.5. Statistical Analysis

One-way analysis of variance (ANOVA) and Tukey’s test were performed to make multiple comparisons and assess the differences in the frequency of the pollen deposits on *A. lepigone* moths collected at various times. A Wilcoxon rank-sum test was utilized to (a) evaluate disparities in the mean annual frequency of pollen occurrence on the proboscises of female and male *A. lepigone* moths and (b) compare the mean annual mean frequency of pollen deposits on male, female, and all *A. lepigone* moth proboscises. A chi-square test was applied to examine the differences in the annual rates of pollen deposition on male, female, and all *A. lepigone* moths and the characteristics of the pollen sources. All statistical analyses were conducted in SPSS v. 13.0 [50].

## 5. Conclusions

Understanding the host plant range of adult insect herbivores is crucial for (a) comprehending their pollination service, interactions with plants, and migration patterns and (b) developing efficacious insect pest management strategies. Here, our pollen analysis established the feeding range of Chinese populations of adult *A. lepigone* sampled on Beihuang Island in the center of the Bohai Strait. This zone is a seasonal migration pathway for *A. lepigone*. To the best of our knowledge, the present study is one of the first to describe host plant use in adult *A. lepigone*. We have provided evidence that this moth is a significant nocturnal flower visitor, carrying pollen from 39 or more species across at least 21 families and 31 genera. This suggests that the moth is an excellent candidate to be considered an important pollinator. Our data also demonstrated that adult *A. lepigone* moths migrate northward in the springtime from northern China into the northeastern agricultural region of China and return southward during the autumn. Subsequent studies should aim to determine the importance of this moth species in pollination. The present work nonetheless lays an empirical foundation for future research that will (a) elucidate the process of coevolution between moths and their host plants and (b) develop novel efficacious management practices for the control of major agricultural insect pests.

## Figures and Tables

**Figure 1 plants-12-03406-f001:**
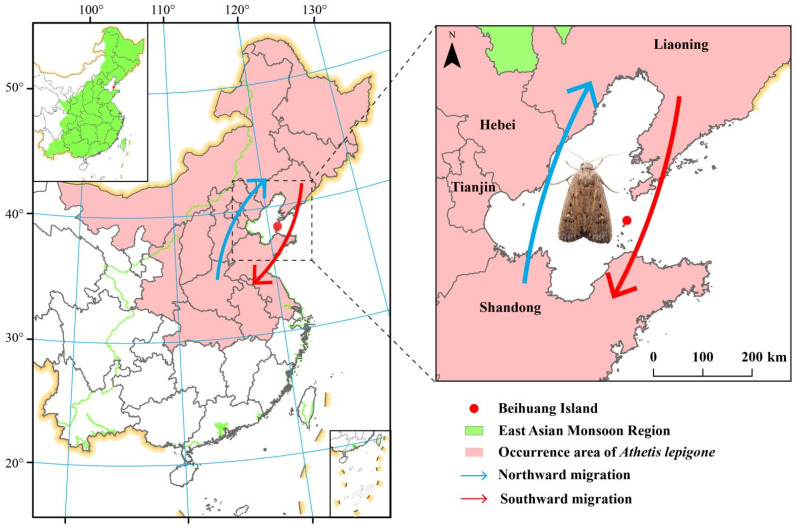
Map showing the occurrence range of *A. lepigone* in China (**left**) and the position of BH Island (38°24′ N, 120°55′ E), the searchlight trap site (**right**), relative to the Bohai Sea and Huanghai Sea (references to [27,28]).

**Figure 2 plants-12-03406-f002:**
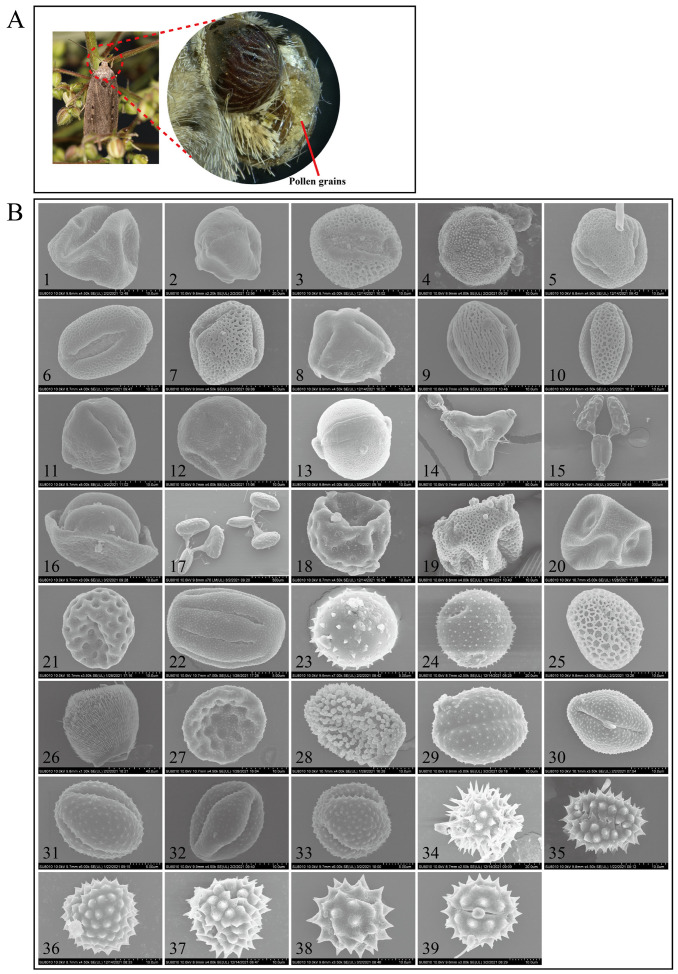
(**A**) Representative image of pollen adhered to the proboscis of a noctuid moth. (**B**) SEM microphotographs of the examined pollen species presented on migratory *Athetis lepigone*. The pollen type numbers in the electron microscope images correspond to the same numbers used in Table 1.

**Figure 3 plants-12-03406-f003:**
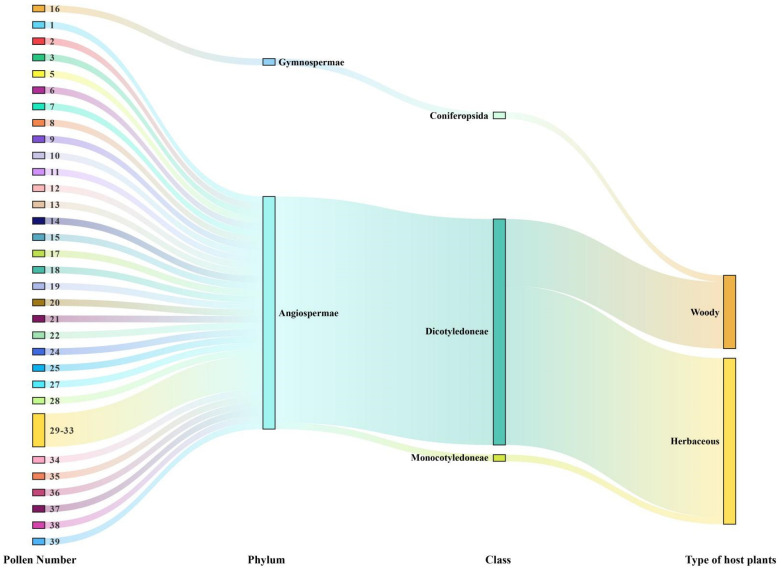
The characteristic of the pollen source plants found on migratory *Athetis lepigone* during 2020–2021.

**Figure 4 plants-12-03406-f004:**
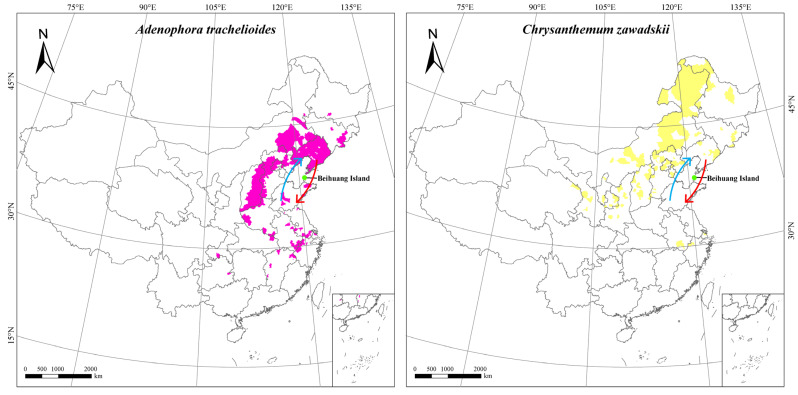
Maps showing the district distribution of *Adenophora trachelioides* (**right**) and *Chrysanthemum zawadskii* (**left**) in China. Different shades represent the plants’ distribution, with arrows indicating the inferred insect migratory paths during different seasons.

**Table 1 plants-12-03406-t001:** Comparative assessment of the degree of taxonomic identification obtained through either molecular or morphology-based approaches, for 39 different types of pollen grains dislocated from *A. lepigone* long-distance migrants collected on Behuang Island (Bohai Sea, Northeastern China). For each type of pollen grain, the highest level of taxonomic identification is indicated and contrasted between molecular and morphology-based approaches.

Number	Identified Plants	Taxonomy (Family)	First Detection Time *	Molecular Identification	Pollen Morphological Traits	
					Size	Ornamentation
1	*Morus alba*	Moraceae	April 18th	*Morus alba*	medium-sized	spiny
2	*Prunus avium*	Rosaceae	April 18th	*Prunus avium*	medium-sized	striate
3	*Salix* L.	Salicaceae	April 18th	Sister to *Salix suchowensis*	small	reticulate
4	Unknown	Unknown	June 18th	Unidentifiable	small	gemmate
5	*Tilia L.*	Malvaceae	June 18th	Sister to *Tilia tuan*	small	reticulate
6	*Amorpha fruticosa*	Fabaceae	June 18th	*Amorpha fruticosa*	small	reticulate
7	*Tamarix chinensis*	Tamaricaceae	June 18th	*Tamarix chinensis*	small	reticulate
8	*Robinia pseudacacia*	Fabaceae	June 19th	*Robinia pseudacacia*	medium-sized	rugulate
9	*Ailanthus altissima*	Simaroubaceae	July 2nd	*Ailanthus altissima*	medium-sized	striate
10	*Brassica* L.	Brassicaceae	July 2nd	Sister to *Brassica juncea/*	medium-sized	reticulate
11	*Torilis arvensis*	Apiaceae	July 2nd	*Torilis arvensis*	medium-sized	rugulate
12	*Sorghum bicolor*	Poaceae	July 17th	*Sorghum bicolor*	medium-sized	spiny
13	*Nicotiana tabacum*	Solanaceae	July 21st	*Nicotiana tabacum*	medium-sized	reticulate
14	*Oenothera* L.	Onagraceae	July 28th	Sister to *Oenothera biennis*	medium-sized	smooth
15	*Vincetoxicum* L.	Apocynaceae	July 28th	Sister to *Vincetoxicum hirundinaria*	very large	smooth
16	*Pinus* L.	Pinaceae	July 28th	Sister to *Pinus thunbergii*	large	cerebroid
17	*Cynanchum acutum*	Apocynaceae	August 11th	*Cynanchum acutum*	very large	smooth
18	*Amaranthus blitum*	Amaranthaceae	August 28th	*Amaranthus blitum*	small	granulate
19	*Cuscuta japonica*	Convolvulaceae	August 30th	*Cuscuta japonica*	medium-sized	reticulate
20	*Humulus lupulus*	Cannabaceae	September 4th	*Humulus lupulus*	small	spiny
21	*Chenopodium album*	Amaranthaceae	September 5th	*Chenopodium album*	medium-sized	granulate
22	*Rubia cordifolia*	Rubiaceae	September 5th	*Rubia cordifolia*	small	granulate
23	Unknown	Unknown	September 6th	Unidentifiable	small	spiny
24	*Adenophora trachelioides*	Campanulaceae	September 9th	*Adenophora trachelioides*	medium-sized	spiny
25	*Oleaceae*	Oleaceae	September 10th	Unidentifiable	small	reticulate
26	Unknown	Unkown	September 11th	Unidentifiable	large	spiny
27	*Suaeda glauca*	Amaranthaceae	September 18th	*Suaeda glauca*	small	granulate
28	Aquifoliaceae	Aquifoliaceae	September 20th	Unidentifiable	small	gemmate
29	*Artemisia* L.	Asteraceae	August 31st	Sister to *Artemisia japonica*	small	spiny
30	*Artemisia* L.	Asteraceae	September 5th	Sister to *Artemisia sieversiana*	small	spiny
31	*Artemisia* L.	Asteraceae	September 9th	Sister to *Artemisia annua*	small	spiny
32	*Artemisia* L.	Asteraceae	September 10th	Sister to *Artemisia scoparia*	small	spiny
33	*Artemisia* L.	Asteraceae	September 13th	Sister to *Artemisia* sp. AD-H	small	spiny
34	*Helianthus annuus*	Asteraceae	September 2nd	*Helianthus annuus*	medium-sized	spiny
35	*Tripolium vulgare*	Asteraceae	September 4th	*Tripolium vulgare*	medium-sized	spiny
36	*Ambrosia trifida*	Asteraceae	September 9th	*Ambrosia trifida*	small	spiny
37	*Aster tataricus*	Asteraceae	September 9th	*Aster tataricus*	medium-sized	spiny
38	*Eclipta prostrata*	Asteraceae	October 18th	*Eclipta prostrata*	medium-sized	spiny
39	*Chrysanthemum indicum*	Asteraceae	October 18th	*Chrysanthemum indicum*	small	spiny

Note: small: 10–25 µm; medium-sized: 26–50 µm; large: 50–100 µm; very large: >100 µm. * indicates the earliest recorded time when the specific pollen type was first observed on an insect.

**Table 2 plants-12-03406-t002:** Results of chi-squared test to compare frequencies of pollen grain attachment by year among male and female moths of *Athetis lepigone*.

Year	Pollen Grain Detection Rate (%)		Chi-Squared Test	
	Female	Male	χ2	*p*
2020	21.35 (117/548)	17.91 (118/659)	2.264	0.1324
2021	38.57 (86/223)	11.79 (52/441)	64.4854	9.72 × 10^−16^
2020–2021	26.33 (203/771)	15.45 (170/1100)	33.5854	6.82 × 10^−9^

Note: All data presented in this table are exclusively from pollen grains found on the proboscis of the insects.

## Data Availability

Not applicable.

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
