# Peer review of "Regional Pollination Activity by Moth Migration in Athetis lepigone"

_plants, 2023, doi:10.3390/plants12193406_

Round 1

Reviewer 1 Report

In this manuscript (plants-2619449) entitled "Regional Pollination Activity by Moth Migration in Athetis lepigone" submitted to Plants, Huiru Jia and colleagues characterized the host plant feeding and pollination ranges of an agriculturally important nocturnal moth species Athetis lepigone by identifying the pollen species adhering to their bodies during long-distance migration. Overall, the findings of the present work provide valuable information on the pollination behavior, geographical origins, and pollination regions of A. lepigone moths and could facilitate the design and optimization of efficacious local and regional management strategies for this important insect. The data are convincing and the writing is clear and straightforward. However, some issues need to be addressed for improving the quality of this manuscript.

1, For Figure 1, represented pictures to show pollen adhering to proboscises of A. lepigone moths should be included in the revision.

2, For Figure 2, panels 14, 15 and 17 should be replaced with magnified pictures to show more pollen details.

3, For Table 1, pollen numbers and/or percentage should be statistically analyzed and included in the revision.  

Author Response

Dear Reviewer 1,

Thank you for taking the time to review our manuscript "Regional Pollination Activity by Moth Migration in Athetis lepigone" and for the valuable feedback you've provided. Your insights greatly assist in enhancing the overall quality of our work.

In response to your comments:

In this manuscript (plants-2619449) entitled "Regional Pollination Activity by Moth Migration in Athetis lepigone" submitted to Plants, Huiru Jia and colleagues characterized the host plant feeding and pollination ranges of an agriculturally important nocturnal moth species Athetis lepigone by identifying the pollen species adhering to their bodies during long-distance migration. Overall, the findings of the present work provide valuable information on the pollination behavior, geographical origins, and pollination regions of A. lepigone moths and could facilitate the design and optimization of efficacious local and regional management strategies for this important insect. The data are convincing and the writing is clear and straightforward. However, some issues need to be addressed for improving the quality of this manuscript.

Response: Thank you for your constructive feedback. Here's our response to each point:

1, For Figure 1, represented pictures to show pollen adhering to proboscises of A. lepigone moths should be included in the revision.

Response: We appreciate your suggestion about the inclusion of pictures to show pollen adhering to the proboscises of A. lepigone moths. Following your feedback, we have incorporated these images in the revised version of the manuscript.

2, For Figure 2, panels 14, 15 and 17 should be replaced with magnified pictures to show more pollen details.

Response: We acknowledge your concerns about the clarity of panels 14, 15, and 17 in Figure 2. We recognize the importance of providing clear and detailed representations. Unfortunately, the current images represent the highest clarity and magnification achievable with our equipment. While we will endeavor to obtain clearer images for future works, it's noteworthy to mention that the clarity of these panels is consistent with other panels in Figure 2. We truly appreciate your understanding and constructive feedback.

3, For Table 1, pollen numbers and/or percentage should be statistically analyzed and included in the revision.  

Response: Your point on the statistical analysis of pollen numbers in Table 1 is well-taken. To clarify, the pollen carriage rates' statistical analysis can be found in Table 2. Our intent was for Table 1 to complement Figure 2 by detailing pollen types and their morphological features. We realize that our original submission might have lacked clarity in presenting this link. In our revised manuscript, we have made efforts to provide a clearer connection between the figures and tables.

We genuinely appreciate your expert review, which has been instrumental in improving our manuscript. We hope that our revisions now meet the standards and expectations of the journal.

Warm regards,

Kongming Wu

Reviewer 2 Report

The authors present an interesting study, looking for a moth's migratory routes based on the pollen grains they carry. They used an innovative methodology based on an integrative approach combining traditional morphological identification (based on scanning electron microscopy images) and single pollen grains DNA barcoding. 

The paper is easy to read, but the authors should emphasise more the main results on the seasonal migratory origin of A. lepigone (an additional figure could be nice), and quantitative data about observed pollen grains should be provided to better appreciate the relative importance of each species/family/genus identified.

I have provided some comments below. I hope they will help improve the manuscript.

Keywords: Athletes epigone and moth migration. Providing keywords that are not part of the title could be more interesting.

Table 1 "Moths collected time" please homogenise decimal numbers 

"Moths collected time." Please specify what it is. What is the unit

How many pollen grains were observed? How many per pollen taxon? It would be interesting to provide quantitative data to estimate the relative importance of each pollen type and identify potential contamination.

Table 2 Where were localised those pollen grains? Are they data only for proboscis? Please specify.

Figure 3 If it is easy to do, maybe place "Gymnosperm" and " Coniferopsida" on the top. That will reduce the crosses.

Figure 4 It could be nice to indicate your sampling area on the map.

Figure 2, Are the numbers referring to numbers in Table 1? Please specify.

Figure 4, What are A and B ? Letters are missing on the maps. Or use Right and Left on the legend.

L 33 a large part of plants, but not all of them…

L 167 "Regions of the plants pollinated by A. lepigone" You can not say that. You did not analyse any pollination (i.e. viable pollen deposition on the stigma that conducts to seed production). Based on your results, you can only provide a list of flower species visited by A. lepigone. However, as you did not provide quantitative data for pollen grains, we can not appreciate which species could represent contamination rather than indicating a plant really visited by the moth.

L 167 to 183 This part is your main result and a very smart way to use your pollen data to identify geographical moth origins and migratory routes. It would be super nice to provide a nice figure to present those results, maybe with arrows on a map showing the moth's origin according to seasons… It would probably be more appreciated than the current figure 4.

L 203 Did you observe pollen grains on the body or only on the proboscis?

L 211 to 214. In L 289-290, you explain that you focus only on pollen present on proboscis, so how can you explain the differences observed by bodyside differences?

L 230-231 Did you observe pollen grains on the body or only on the proboscis, as mentioned in the material and method section?

L 242 body of proboscis ?

L 256 "evidence for the its" please correct

L 278 "journey" is repeated twice.

L 327 How many pollen grains were sampled and analysed? Did each pollen grain was genetically analysed? Or only a few per morphotype? 

L 359 "pollinator" you provide no evidence for that, but you observed that it is an important nocturnal flower visitor that carries the pollen of…  That moth is so an excellent candidate to be an important pollinator.

There are a few mistakes here and there; a more careful proofreading should suffice to correct these errors.

Author Response

Dear Reviewer 2,

Thank you for your thorough review of our manuscript. Your insightful feedback has significantly helped in refining our work, ensuring its clarity, and enhancing its overall quality. We truly appreciate your expert guidance.

Below, we address each of your comments and suggestions:

#Reviewer 2

The authors present an interesting study, looking for a moth's migratory routes based on the pollen grains they carry. They used an innovative methodology based on an integrative approach combining traditional morphological identification (based on scanning electron microscopy images) and single pollen grains DNA barcoding. The paper is easy to read, but the authors should emphasis more the main results on the seasonal migratory origin of A. lepigone (an additional figure could be nice), and quantitative data about observed pollen grains should be provided to better appreciate the relative importance of each species/family/genus identified. I have provided some comments below. I hope they will help improve the manuscript.

Response: We truly appreciate the time and effort you've put into reviewing our manuscript. Here's how we addressed each of your concerns:

  1. Keywords: Athletes lepigoneand moth migration. Providing keywords that are not part of the title could be more interesting.

Response: We have introduced additional keywords, “ noctuid moth” and “migration routes”, that aren't part of the title for better representation.

  1. Table 1 "Moths collected time" please homogenise decimal numbers

Response: We apologize for the oversight in "Moths collected time". To clarify, it indicates the earliest date a pollen type was detected on an insect. In our revised manuscript, we've changed it to "First Detection Time*" and added a footnote: "*indicates the earliest recorded time when the specific pollen type was first observed on an insect."

  1. "Moths collected time." Please specify what it is. What is the unit

Response: As mentioned above, we have specified the unit of measurement for better clarity.

  1. How many pollen grains were observed? How many per pollen taxon? It would be interesting to provide quantitative data to estimate the relative importance of each pollen type and identify potential contamination.

Response: We wholeheartedly agree on the importance of providing quantitative data to estimate the relative significance of each pollen type and to identify potential contamination. However, as illustrated in revised Figure 2, due to the large amount of pollen carried by each moth and the tendency for pollen grains to clump together, obtaining precise counts for individual specimens is challenging. Therefore, we couldn't provide such data in this study. In the future, we will explore methods to obtain this critical data.

  1. Table 2 Where were localised those pollen grains? Are they data only for proboscis? Please specify.

Response:The data in Table 2 pertains exclusively to pollen grains found on the proboscis. We've also added a note to the table's caption for clarity: “Note: All data presented in this table are exclusively from pollen grains found on the proboscis of the insects.”

  1. Figure 3 If it is easy to do, maybe place "Gymnosperm" and " Coniferopsida" on the top. That will reduce the crosses.

Response: As you suggested, "Gymnosperm" and "Coniferopsida" have been positioned at the top to enhance the figure's clarity.

  1. Figure 4 It could be nice to indicate your sampling area on the map.

Response: The sampling area is now clearly indicated on the map.

  1. Figure 2, Are the numbers referring to numbers in Table 1? Please specify.

Response: We've clarified the correlation between the numbers in this figure and those in Table 1.

  1. Figure 4, What are A and B ? Letters are missing on the maps. Or use Right and Left on the legend.

Response: We've updated the legend in Figure 4 for better clarity, now using "Right" and "Left" labels instead.

  1. L 33 a large part of plants, but not all of them…

Response: The phrasing on L 33 has been adjusted to reflect a more precise statement.

  1. L 167 "Regions of the plants pollinated by A. lepigone" You can not say that. You did not analyse any pollination (i.e. viable pollen deposition on the stigma that conducts to seed production). Based on your results, you can only provide a list of flower species visited by lepigone. However, as you did not provide quantitative data for pollen grains, we can not appreciate which species could represent contamination rather than indicating a plant really visited by the moth.

Response: Thank you for highlighting this distinction. We realize our phrasing may have inadvertently suggested an analysis of the pollination process, which was not the case. In fact, our data only allows us to infer the flower species visited by A. lepigone. To address this, we have revised the original text from "Regions of the plants pollinated by A. lepigone" to "Possible migration trajectories of A. lepigone". Your feedback is invaluable, emphasizing the need for a clear delineation between visitation and actual pollination in our descriptions and future research.

  1. L 167 to 183 This part is your main result and a very smart way to use your pollen data to identify geographical moth origins and migratory routes. It would be super nice to provide a nice figure to present those results, maybe with arrows on a map showing the moth's origin according to seasons… It would probably be more appreciated than the current figure 4.

Response: As you suggested, we've introduced a new figure to better visualize the moth's origin according to seasons.

  1. L 203 Did you observe pollen grains on the body or only on the proboscis?

Response: We did observe pollen grains on other parts of the body, but these observations were not systematically recorded. As pointed out in our manuscript, numerous studies have indicated that the proboscis is the primary site for pollen carriage. To enhance the efficiency of our work, we focused primarily on the proboscis, conducting only preliminary inspections on other body parts without formal recording. As mentioned in our materials and methods section, “the present study mainly focused on the proboscises of the A. lepigone moths sampled.”

  1. L 211 to 214. In L 289-290, you explain that you focus only on pollen present on proboscis, so how can you explain the differences observed by bodyside differences?

Response: Thank you for pointing out this inconsistency. As stated in our materials and methods section, our primary focus was on the pollen found on the proboscises of the A. lepigone moths sampled. The observed differences by body side mainly stem from our preliminary observations. We apologize for any confusion and have made the necessary clarifications in the revised manuscript.

  1. L 230-231 Did you observe pollen grains on the body or only on the proboscis, as mentioned in the material and method section?

Response: Yes, we did observe pollen grains on the body. However, these observations were not formally recorded. As mentioned, all data presented in this paper are derived from observations on the proboscis. We have added a note in the manuscript to clarify this point.

  1. L 242 body of proboscis ?

Response: Thank you for pointing that out. The term "body" on L 242 has been revised to "proboscis" for clarity and accuracy.

  1. L 256 "evidence for the its" please correct. L 278 "journey" is repeated twice.

Response: The grammatical issues on L 256 and L 278 have been corrected.

  1. L 327 How many pollen grains were sampled and analysed? Did each pollen grain was genetically analysed? Or only a few per morphotype?

Response: All the pollen grains we obtained were photographed and subjected to molecular identification. After processing and categorizing, we presented the results in the current paper. If necessary, we can provide all the original photographs as supplementary material for reference. 

  1. L 359 "pollinator" you provide no evidence for that, but you observed that it is an important nocturnal flower visitor that carries the pollen of…  That moth is so an excellent candidate to be an important pollinator.

Response: Thank you for your suggestion. We have revised the inappropriate wording in the original text to read: “ We have provided evidence that this moth is a significant nocturnal flower visitor, carrying pollen from 39 or more species across at least 21 families and 31 genera. This suggests that the moth is an excellent candidate to be considered an important pollinator.”

Once again, thank you for your meticulous review. We sincerely hope that our revisions now align with the journal's standards and address all your concerns.

Warm regards,

Kongming Wu

Author Response

Dear Reviewer,

We greatly appreciate your meticulous attention to detail in reviewing our manuscript. Your formatting suggestions and detailed revisions have been invaluable. We have carefully gone through each of your comments and have incorporated all the suggested changes. The improved readability and presentation of our work, thanks to your recommendations, will undoubtedly benefit the readers.

Thank you for the time and effort you have invested in refining our manuscript. Your expertise and keen eye for detail have been instrumental in enhancing the overall quality of our paper.

Warm regards,

Kongming Wu

Reviewer 4 Report

“Regional Pollination Activity by Moth Migration in Athetis lepigone” presents an interesting and relevant topic. However, there are major flaws that need to be addressed, particularly in the introduction and discussion of results. Please see comments below.

 The introduction needs to be restructured. It is somewhat confusing, lacks coherent flow of ideas and focus. If the focus is pollination, it should be focused on that. Mixing pollinating adults with herbivorous behaviour of the larvae makes the speech confusing. From L32 to L62 there is a mix of different topic without a logical flow. After reading the discussion, most of the information in L189 to 228 could be used to build the introduction.  

 L81-82: This sentence makes no sense. It is likely incomplete.

 L103-105: I cannot understand how this sentence is relevant in the results section. There is no discussion of pollen morphological traits or its link to other results in the study. Perhaps it is better to include this in the methods.

 L144: replace “thirty” with “30”

 L181-183: During the autumn migration period, however, Chrysanthemum zawadskii and Adenophora trachelioides pollen grains were detected on A. lepigone. For this reason, these migrants were distributed mainly in Northeast China (Figure 4).” I cannot understand the emphasis given to these species. Background is missing from the introduction and reasons that could explain the relevance of this result are missing from the discussion.

L230-232: This sentence should not be in the discussion. It could make sense in the introduction, followed by a justification for why it is relevant to study this species. For example, L63 to 75 could provide such justification.

  L236-237 “However, we propose that the actual scope of plant species visited by A. lepigone exceeded those enumerated in this investigation.” Why do you propose this? You could present arguments to support this statement.

L238-242 “the host plants of A. lepigone have strong geographical specificity. We found that nearly all pollen taxa detected were borne by one or more moth species regardless of the number of moths examined. Thus, there is a certain degree of homogeneity in terms of the types of flowers pollinated by nocturnal moths. Pollen grains from Oenothera L. and Vincetoxicum L. were found on the bodies of almost all moth species explored here.” Your data does not support this. You refer several moth species, but your data focuses on only 1 species.

 Overall, in the discussion section you do not discuss your results at all.

 L278: The word “journey” is duplicated.

Please see comments and suggestions.

Author Response

Dear Reviewer,

Thank you for your thorough feedback. Here are the changes we've made based on your comments:

  1. The introduction needs to be restructured. It is somewhat confusing, lacks coherent flow of ideas and focus. If the focus is pollination, it should be focused on that. Mixing pollinating adults with herbivorous behaviour of the larvae makes the speech confusing. From L32 to L62 there is a mix of different topic without a logical flow. After reading the discussion, most of the information in L189 to 228 could be used to build the introduction.   

Response: We have restructured the introduction to improve its coherence and flow, emphasizing the focus on pollination. The herbivorous behavior of larvae has been separated from the pollinating role of the adults to avoid confusion. We've incorporated information from L189 to 228, which was previously in the discussion, to provide better background and context.

  1. L81-82: This sentence makes no sense. It is likely incomplete.

Response: We apologize for the oversight. We have revised the sentence to make it coherent and relevant.

  1. L103-105: I cannot understand how this sentence is relevant in the results section. There is no discussion of pollen morphological traits or its link to other results in the study. Perhaps it is better to include this in the methods.

Response: We understand your point. However, we'd like to emphasize that morphological identification is vital to our study. Criteria like size and surface structure, which have been highlighted in the methods section, are crucial for this purpose. Therefore, we deemed it necessary to include a descriptive portion in the results section.

  1. L144: replace “thirty” with “30”

Response: The word “thirty” has been changed to the numeral “30” as per your recommendation.

  1. L181-183: “During the autumn migration period, however, Chrysanthemum zawadskii and Adenophora trachelioides pollen grains were detected on A. lepigone. For this reason, these migrants were distributed mainly in Northeast China (Figure 4).” I cannot understand the emphasis given to these species. Background is missing from the introduction and reasons that could explain the relevance of this result are missing from the discussion.

Response: Pollen analysis is a widely used tool in analyzing insect dietary patterns and migration routes. Recognizing pollen on an insect aids in identifying potential migration sources. The geographical occurrence of Chrysanthemum zawadskii and Adenophora trachelioides, as we discussed, provides substantial proof of our migration conclusions. Although the introduction might not provide extensive background on this, we believe our results and discussion make the relevance evident. Still, we appreciate your input and will amend our text for better reader comprehension.

  1. L230-232: This sentence should not be in the discussion. It could make sense in the introduction, followed by a justification for why it is relevant to study this species. For example, L63 to 75 could provide such justification.

Response: Per your suggestion, we’ve moved this statement to the introduction as suggested. In conjunction with L63 to 75, it now serves to emphasize the importance of studying A. lepigone's role in pollination.

  1. L236-237 “However, we propose that the actual scope of plant species visited by lepigone exceeded those enumerated in this investigation.” Why do you propose this? You could present arguments to support this statement.

Response: Our intention was not conveyed accurately. We have revised it to:“It's important to note that our samples were exclusively sourced from a single location. Considering the inherent geographical specificity in plant distribution[21], we are confident that the true count of plant species frequented by A. lepigone surpasses what we've documented in this study.”

  1. L238-242 “the host plants of A. lepigone have strong geographical specificity. We found that nearly all pollen taxa detected were borne by one or more moth species regardless of the number of moths examined. Thus, there is a certain degree of homogeneity in terms of the types of flowers pollinated by nocturnal moths. Pollen grains from Oenothera L. and Vincetoxicum L. were found on the bodies of almost all moth species explored here.” Your data does not support this. You refer several moth species, but your data focuses on only 1 species.

Response: We regret the oversight. The reference to various moth species was an unintended error. We've adjusted the content to clarify that our study strictly concerns A. lepigone and have removed any generalized claims about nocturnal moths to maintain accuracy.

  1. L278: The word “journey” is duplicated.

Response: We're sorry for this mistake. The redundancy in the use of the word “journey” has been addressed.

We are genuinely grateful for your insightful comments. They have undeniably enhanced our manuscript's coherence and quality. We are optimistic that these revisions adequately tackle your reservations and refine the manuscript accordingly.

Warm regards,

Kongming Wu

Round 2

Reviewer 4 Report

The authors have addressed the comments. I have only a few minor comments.

L223-25 The "," before "the present work" should be replaced ".", otherwise the sentence does not make sense.

L233-34 There is still reference to multiple moth species. You may have missed this one.

See comments.

Author Response

Dear Reviewer 4#,

Thank you for your further feedback after our revisions. We appreciate the additional time you have taken to guide us towards a more polished manuscript. We have addressed the minor comments you raised in this second round of review as follows:

Reviewers 4#

Comments and Suggestions for Authors:

The authors have addressed the comments. I have only a few minor comments.

  1. L223-25 The "," before "the present work" should be replaced ".", otherwise the sentence does not make sense.

Response: Thank you for pointing out the punctuation error on Lines 223-25. We have corrected it by replacing the comma with a period before 'the present work' to enhance sentence clarity.

  1. L233-34 . There is still reference to multiple moth species. You may have missed this one.

Response: Thank you for highlighting this oversight. Indeed, this was an area we had addressed in our previous revision but missed on Line L233-34. We have now added the appropriate references to this section to support our mention of multiple moth species. We greatly appreciate your thorough review, which ensures the quality and accuracy of our paper.

We are grateful for your meticulous reviews and constructive suggestions, and we believe these further modifications will enhance the quality of our manuscript.

Warm regards,

Kongming Wu